# Asymmetric thinning of the cerebral cortex across the adult lifespan is accelerated in Alzheimer's disease

James M. Roe[1✉], Didac Vidal-Piñeiro [1], Øystein Sørensen [1], Andreas M. Brandmaier [2], Sandra Düzel[2], Hector A. Gonzalez[3], Rogier A. Kievit [4], Ethan Knights[4], Simone Kühn[2,5], Ulman Lindenberger [2,6], Athanasia M. Mowinckel[1], Lars Nyberg[7], Denise C. Park[3], Sara Pudas [7], Melissa M. Rundle[3], Kristine B. Walhovd [1,8], Anders M. Fjell[1,8], René Westerhausen [1] & The Australian Imaging Biomarkers and Lifestyle Flagship Study of Ageing*

Aging and Alzheimer's disease (AD) are associated with progressive brain disorganization. Although structural asymmetry is an organizing feature of the cerebral cortex it is unknown whether continuous age- and AD-related cortical degradation alters cortical asymmetry. Here, in multiple longitudinal adult lifespan cohorts we show that higher-order cortical regions exhibiting pronounced asymmetry at age ~20 also show progressive asymmetry-loss across the adult lifespan. Hence, accelerated thinning of the (previously) thicker homotopic hemisphere is a feature of aging. This organizational principle showed high consistency across cohorts in the Lifebrain consortium, and both the topological patterns and temporal dynamics of asymmetry-loss were markedly similar across replicating samples. Asymmetry-change was further accelerated in AD. Results suggest a system-wide dedifferentiation of the adaptive asymmetric organization of heteromodal cortex in aging and AD.

[1] Center for Lifespan Changes in Brain and Cognition, Department of Psychology, University of Oslo, Oslo, Norway. [2] Center for Lifespan Psychology, Max Planck Institute for Human Development, Berlin, Germany. [3] Center for Vital Longevity, University of Texas, Dallas, TX, USA. [4] MRC Cognition and Brain Sciences Unit, University of Cambridge, Cambridge, UK. [5] Department of Psychiatry and Psychotherapy, University Medical Center Hamburg-Eppendorf, Hamburg, Germany. [6] Max Planck UCL Centre for Computational Psychiatry and Ageing Research, Berlin, Germany. [7] Umeå Center for Functional Brain Imaging and Department of Integrative Medical Biology, Umeå University, Umeå, Sweden. [8] Department of Radiology and Nuclear Medicine, Oslo University Hospital, Oslo, Norway. *A list of authors and their affiliations appears at the end of the paper. ✉email: j.m.roe@psykologi.uio.no

Healthy aging and Alzheimer's disease (AD) are associated with progressive disruption of structural brain organization[1,2] and the dedifferentiation of functionally specialized systems[3–7]. Despite being a global organizing property of the cortex[8,9] with plausible relevance for hemispheric specialization[10], cortical thickness asymmetry has been mostly overlooked in studies examining cortical aging. Yet an asymmetrically organized cortex gives rise to efficient functional network organization[11–14] and thus may support cognition[15,16]. Hence, as the cortex thins over time[17], cortical thickness asymmetry may also change, which may be informative for declining brain function. Here, utilizing large longitudinal data sets from five healthy aging cohorts and one dementia cohort, we aimed to establish the trajectories of asymmetric cortical thinning and examine deviations from these trajectories in AD patients.

Of the few cross-sectional studies assessing age effects upon cortical thickness asymmetry, reported results have been inconsistent[8,15,18,19]. For example, in prefrontal regions known to be especially vulnerable in aging[17,20], a steeper relationship for thickness with age has been reported in both the right hemisphere (RH)[18] and left hemisphere (LH)[15] between ages 5 and 60. Still, recent large-scale meta-analyses in the ENIGMA consortium found no evidence for age effects upon frontal asymmetry[8], and studies have linked age to both loss[19] and widespread exacerbation of[18] thickness asymmetries established earlier in life. Conflicting results may be partly ascribable to differences in sample size and test power, and linear modeling of cross-sectional age effects across developmental and aging samples known to follow nonlinear brain trajectories over time[21,22]. Methodological differences such as whether age effects upon thickness asymmetry were probed regionally[8,19] or vertex-wise across the cortex[15,18] may also help explain divergent results. Longitudinal lifespan studies examining intraindividual asymmetry change have the potential to resolve inconsistencies and capture nonlinear asymmetry-change trajectories, but are currently absent from the literature. Thus, the foundational question of whether and where the cerebral hemispheres thin at different rates in aging remains open.

Regions of cortex that thin most in healthy aging overlap with regions exhibiting more degeneration in AD, indicating accelerated change trajectories in AD[23–25], and that normal and pathological brain aging may at least partly exist on a continuum[26]. Thus, hypothesized changes in asymmetry across healthy adult life may be evident to a greater extent in AD. Although cross-sectional findings are somewhat conflicting[19,27,28] and meta-analyses report scant evidence for increased LH vulnerability in AD[29], early longitudinal findings indicate that disease progression may be associated with faster LH degeneration in medial and prefrontal cortex[30]. Furthermore, recent evidence hints that system-wide loss of existing asymmetries may be part of the AD phenotype[31]. These results suggest that altered cortical asymmetry may accompany increased intraindividual cognitive and clinical deficit in AD over time, supporting a role for structural asymmetry in healthy brain function. Currently, however, it is not known whether purported AD-related alterations in cortical asymmetry reflect an acceleration of gradual changes occurring over the healthy adult lifespan.

Here, in an initial longitudinal adult lifespan discovery sample (Center for Lifespan Changes in Brain and Cognition, LCBC; see Fig. 1a; age range = 20–89; total observations = 2577; longitudinal observations = 1851; mean follow-up interval = 2.7 years), we show that the cerebral cortex thins asymmetrically in aging. We then sought to replicate the findings in four independent longitudinal adult lifespan samples (see Fig. 1b and Supplementary Table 1). To capture nonlinear differences between homotopic thinning profiles across adult life, we performed vertex-wise Generalized Additive Mixed Models (GAMMs)[32] after resampling individual LH and RH thickness maps to a common surface[33,34] (Fig. 1c and Supplementary Fig. 1). Next, we applied data-driven clustering to identify regions of cortex with similar profiles of changing thickness asymmetry across adult life (Fig. 1d, e). We hypothesized that cortical regions that are strongly asymmetric in young adulthood—as a plausible structural marker of hemispheric specialization reflecting optimal brain organization[10]—will exhibit changing asymmetry in aging. Finally, we predicted that similar changes in regional thickness asymmetry will occur faster in AD and show this using a longitudinal clinical dementia data set.

## Results

**Asymmetry-change analyses.** To identify cortical regions changing in thickness asymmetry across adult life, we ran vertex-wise GAMMs with Age × Hemisphere (i.e., age-related change in asymmetry) as the effect of interest in the LCBC discovery sample. Significant age-related changes in asymmetry were found in large portions of prefrontal, anterior/middle temporal, insular, and lateral parietal cortex (Fig. 2a), with particularly pronounced effects in medial prefrontal cortex (mPFC). The main effect of Hemisphere (i.e., irrespective of Age) revealed a clear anterior–posterior pattern of general thickness asymmetry in adult life (Fig. 2b), consistent with a recent meta-analysis[8]. Critically, Age × Hemisphere effects conformed well to the main effect of Hemisphere, indicating that regions of cortex characterized by strong asymmetry also exhibit changing thickness asymmetry with age. Note that the direction of asymmetry-change effects is not interpretable from the GAMM interaction (see next section). An equivalent analysis with a linear mixed-effect (LME) model showed comparable results (see Supplementary Note 1 and Supplementary Fig. 2).

**Clustering of asymmetry trajectories.** To explore asymmetry trajectories underlying the Age × Hemisphere GAMM interaction, we clustered the cortex to identify regions exhibiting similar profiles of changing thickness asymmetry across the adult lifespan. PAM clustering[36] indicated that asymmetry trajectories were optimally partitioned using a three-cluster solution (mean silhouette width = 0.46; Fig. 1e). Figure 3a–c shows the resulting three-cluster partition. Cluster 1 consisted of vertices exhibiting mean loss of leftward asymmetry with age (negative slope from positive intercept) and mapped onto medial, orbitofrontal, dorsolateral frontal, and anterior temporal cortex. Conversely, Cluster 3 vertices exhibited mean loss of rightward asymmetry with age (positive slope from negative intercept), and mapped predominantly onto insular, lateral temporoparietal cortex, and caudal anterior cingulate. Cluster 1 and Cluster 3 trajectories showed asymmetry declines emerging at a similar age (around early 30s) and continuing across the adult lifespan, with accelerated decline around age 60 (see also Supplementary Fig. 2D). Total loss of both leftward and rightward asymmetry occurred at a similar age, around the mid 70s. Beyond this, a continuing trend toward the opposite asymmetry to youth was apparent, particularly in Cluster 1. Cluster 2 trajectories were more mixed and generally showed weaker effects, although a trend toward loss of leftward asymmetry was still apparent. These results were robust to varying the number of clustering partitions (Supplementary Fig. 3). Homotopic age trajectories are shown in Fig. 3d for eight main clustering-derived regions of interest (ROI's) selected based on cluster- and effect-size criteria (see also Supplementary Fig. 4). Leftward asymmetry loss in frontal and anterior temporal cortex was driven by accelerated LH-thinning. Conversely, rightward asymmetry loss was driven by accelerated RH-thinning (e.g., in

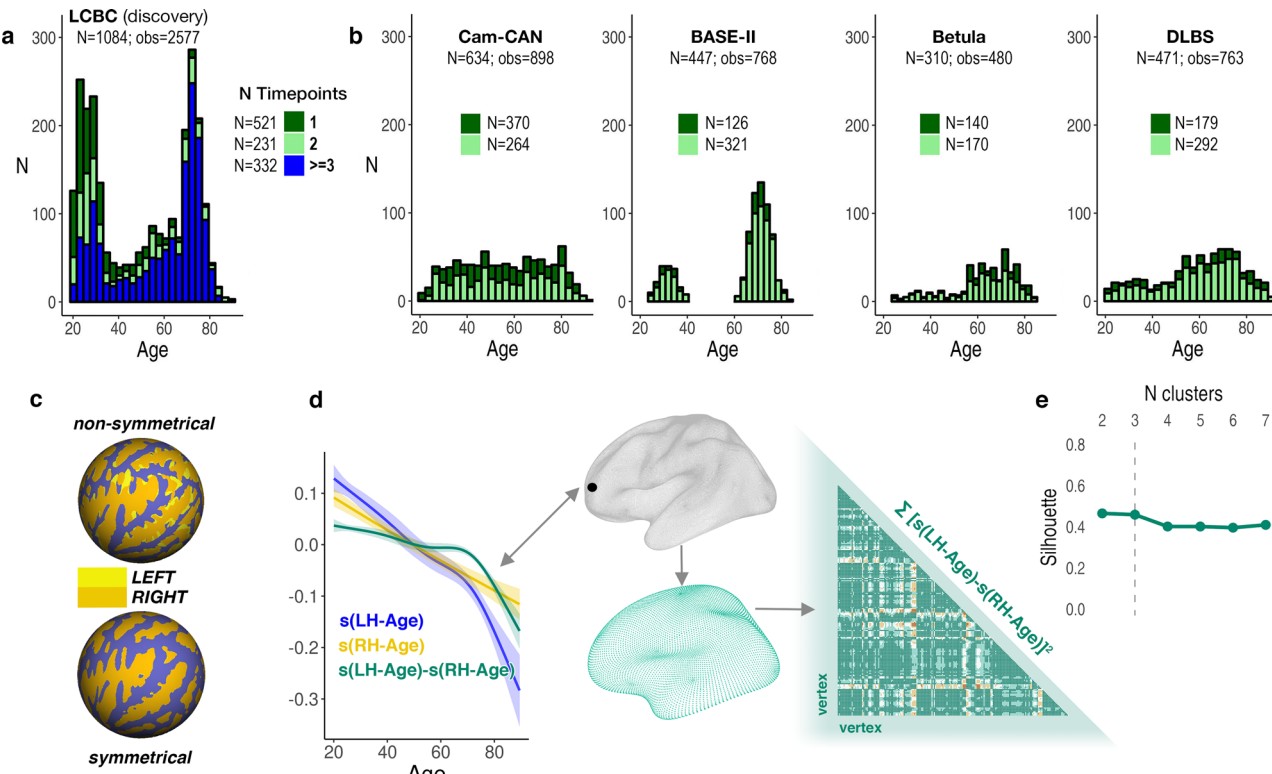

**Fig. 1 Samples and analysis method. a** Age-distribution of the LCBC discovery sample. **b** Age-distributions of the four replication samples. Colors in (**a**, **b**) highlight the number of longitudinal scans available per sample (i.e., dark green denotes subjects with only one timepoint, light green denotes subjects with two timepoints, blue denotes subjects with more than two timepoints; stacked bars). The number of unique subjects ($N$) is given in total and per number of available timepoints. The total number of scan observations (obs) is given per cohort. **c** Spherical surface rendering of a nonsymmetrical (top) compared to the symmetrical template (bottom) used in the present study. Mean sulcal topography of the left and right hemispheres is shown in yellow and orange, respectively. Note the near-perfect alignment of left- and right-hemisphere topography on the symmetrical surface. **d** Visualization of asymmetry analysis at an example vertex (black circle). GAMMs were used to compute the zero-centered (i.e., demeaned) age trajectories of the left [s(LH-Age)] and right [s(RH-Age)] hemisphere at every vertex. Blue and yellow lines indicate the smooth age trajectory of mean cortical thickness for the left and right hemisphere, respectively, after each has been demeaned. The difference between these (green line) depicts the asymmetry trajectory [s(LH-Age)-s(RH-Age)] that was used for assessment of age-change in asymmetry, equivalent to the GAMM interaction term. Ribbons depict SEM. Significant asymmetry trajectories were downsampled to a lower resolution template (fsaverage5; $N$ vertices = 10,242) and a dissimilarity matrix based on the distance between trajectories at every vertex pair (computed by sum of least squares) was obtained and submitted to partition around medoids (PAM) clustering. **e** Mean silhouette width was used to determine the optimal partition number. LH left hemisphere, RH right hemisphere.

insular and lateral parietal cortex; Fig. 3d). Only the cingulate region did not conform to this pattern, exhibiting an early leftward asymmetry that increased with age. Importantly, accelerated thinning corresponded to the percentage of cortex lost with age

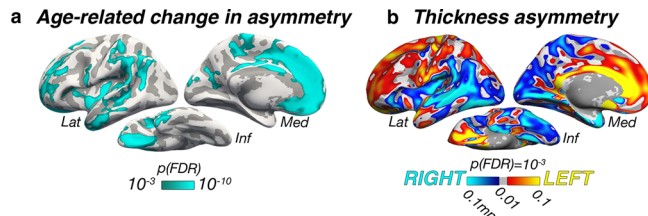

**Fig. 2 Vertex-wise GAMM effects. a** Significance map for age-related changes in asymmetry in the LCBC discovery sample (2577 observations; Age × Hemisphere GAMM interaction; one-sided $F$-test). **b** Mean thickness asymmetry for the discovery sample, irrespective of age (main effect of Hemisphere; two-sided test). Only regions showing significant asymmetry are shown. Warm and cold colors indicate leftward and rightward thickness asymmetry (mm), respectively. Maps in (**a**, **b**) are corrected for multiple comparisons using false discovery rate (FDR)[35] at $p$(FDR) < 0.001. Lat lateral, Med medial, Inf inferior.

(i.e., hemispheric differences in rates of thinning were relative; also indexed by crossing trajectories; Fig. 3d). Overall, findings in the discovery sample indicate that changes in cortical thickness asymmetry are a feature of aging, and that accelerated thinning of the thicker hemisphere leads to extensive loss (or reversal) of asymmetry in older age.

**Replication analyses**. We next tested whether the clustering of asymmetry trajectories was reliable across four independent longitudinal adult lifespan cohorts (Cambridge Center for Aging and Neuroscience (Cam-CAN), Berlin Study of Aging-II (BASE-II), BETULA, and Dallas Lifespan Brain Study (DLBS): described in Supplementary Table 1 and Fig. 1b). Clustering solutions in Cam-CAN, BASE-II, and BETULA exhibited highly similar mean intercept and slope effects to those described in the discovery sample, and mapped onto near-identical regions (Fig. 4). In these, asymmetry-loss effects conformed to the anterior–posterior pattern of general cortical asymmetry in the same manner as in the discovery sample. Quantitatively, the clustering trajectories in Cam-CAN, BASE-II, and BETULA were comparable to the clustering trajectories in LCBC (Supplementary Fig. 5). Spatially, the similarity of clustering results to those observed in the discovery sample was substantiated by Dice coefficients far higher

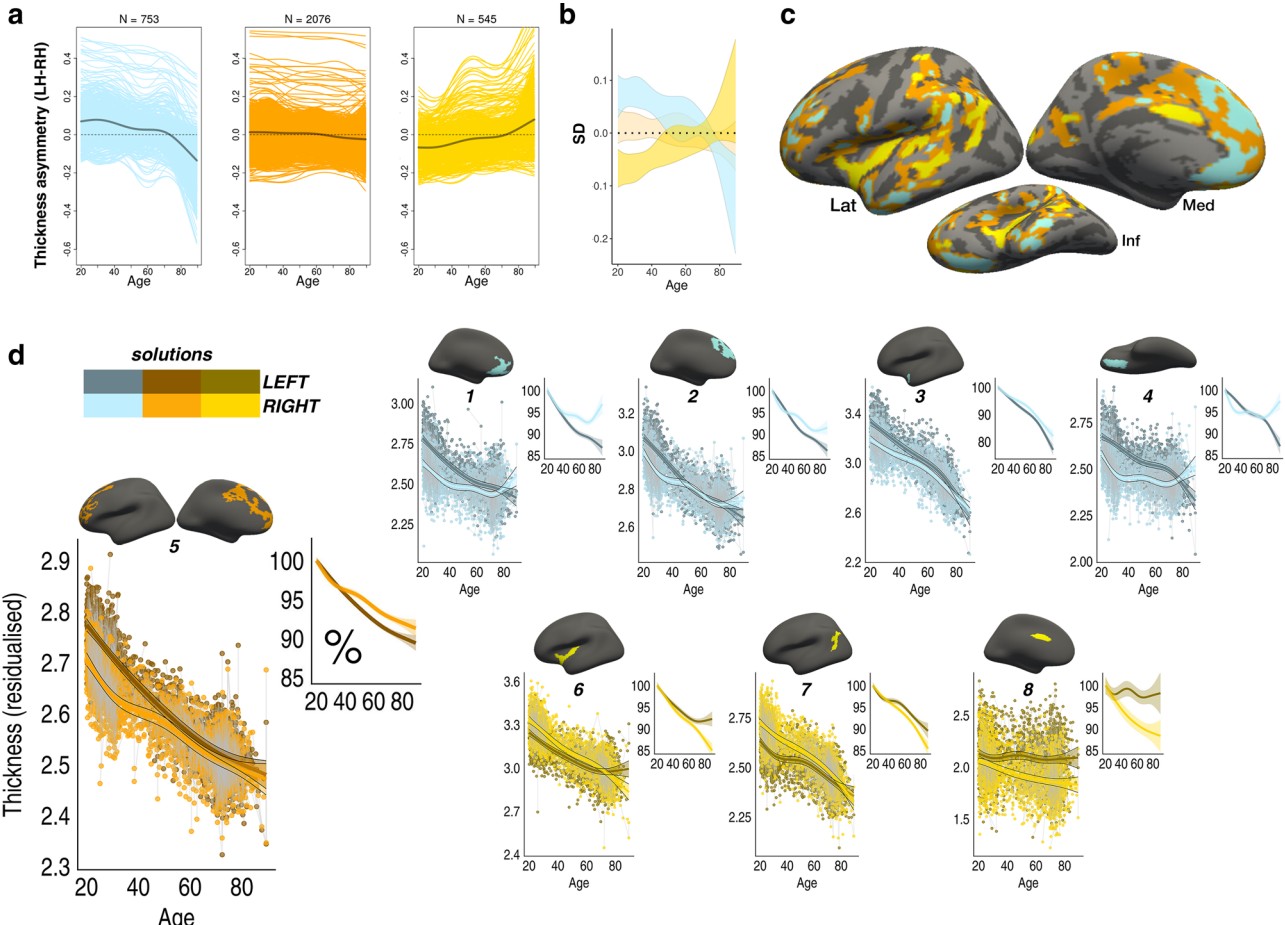

**Fig. 3 Clustered trajectories. a** PAM clustering solutions for asymmetry trajectories in the LCBC discovery sample. Each colored line represents a vertex (N = number of vertices within each solution). Mean trajectories are in gray. Vertices showing leftward asymmetry in early adult life (higher than dotted line) typically exhibit loss of leftward asymmetry with age (blue plot), whereas vertices showing rightward asymmetry (lower than dotted line) typically exhibit loss of rightward asymmetry (yellow plot). Importantly, because asymmetry trajectories were computed as the difference between zero-centered hemispheric trajectories [s(LH-Age)-s(RH-Age)] (cf. Fig. 1d), mean differences between LH and RH thickness (i.e., mean asymmetry/the intercept) are not taken into account and do not influence the clustering. For visualization of the absolute asymmetry trajectories we added the main effect of Hemisphere, vertex-wise, and computed the mean asymmetry trajectory of vertices in each clustering solution. **b** Standard deviations (SD) of the asymmetry trajectories for the clustering solutions. **c** Solutions mapped on the surface. **d** Thinning trajectories plotted separately for LH and RH in regions derived from the clustering (numbered 1–8). Colors correspond to the solutions in (**a**), and darker shades indicate LH trajectories. All trajectories were fitted using GAMMs. Data are residualized for sex, scanner, and random subject intercepts. Trajectories depict mean thickness and ribbons depict 95% confidence intervals. Smaller plots illustrate percentage change with age for each region. As outliers were removed on a region-wise basis (see "Methods"), the number of observations underlying plots 1–8 is: 5150, 5154, 5148, 5152, 5150, 5150, 5148, 5154, respectively. LH left hemisphere, RH right hemisphere, Lat lateral, Med medial, Inf inferior.

than expected by chance (Cam-CAN = 0.54; BASE-II = 0.56; BETULA = 0.50; DLBS = 0.41; true expected Dice at random = 0.1, all $p_{perm} < 0.001$). The clustering in DLBS showed only partial replication: contrary to other samples, mPFC and temporal cortex clustered with regions characterized by mean rightward asymmetry loss, and intercept effects indicative of early leftward asymmetry were not evident in the estimated asymmetry trajectories in DLBS. Vertex-wise GAMM effects also appeared largely consistent across samples (see Supplementary Note 2 and Supplementary Figs. 6 and 7). Overall, the clustering of thickness asymmetry age trajectories in the LCBC discovery sample replicated in three of four longitudinal aging cohorts (see Supplementary Figs. 8 and 9 for full results varying the number of clustering partitions in each cohort).

**Cognitive-change analysis**. We next a ran GAMM for each ROI to assess whether longitudinal thickness asymmetry change

predicted longitudinal cognitive change in verbal memory and fluid reasoning ability—cognitive domains that show high vulnerability in aging[37]. While age explained 29% and 36% of the variance in longitudinal memory and fluid reasoning scores, respectively (sex controlled; Supplementary Fig. 10 and Supplementary Table 4), we observed no significant (pFDR < 0.05) effects of thickness asymmetry change on longitudinal change for either cognitive measure in any of the eight ROIs (Supplementary Table 5). Furthermore, including all eight asymmetry ROI's in a GAMM did not significantly improve model fit, suggesting that asymmetry change had no additive effect upon longitudinal cognitive change (verbal memory $p = 0.86$; fluid reasoning $p = 0.84$; see Supplementary Note 3).

**Longitudinal AD analysis**. Finally, we asked whether in AD, asymmetry loss is accelerated in regions prone to exhibit asymmetry change in healthy aging. We used data from the Australian

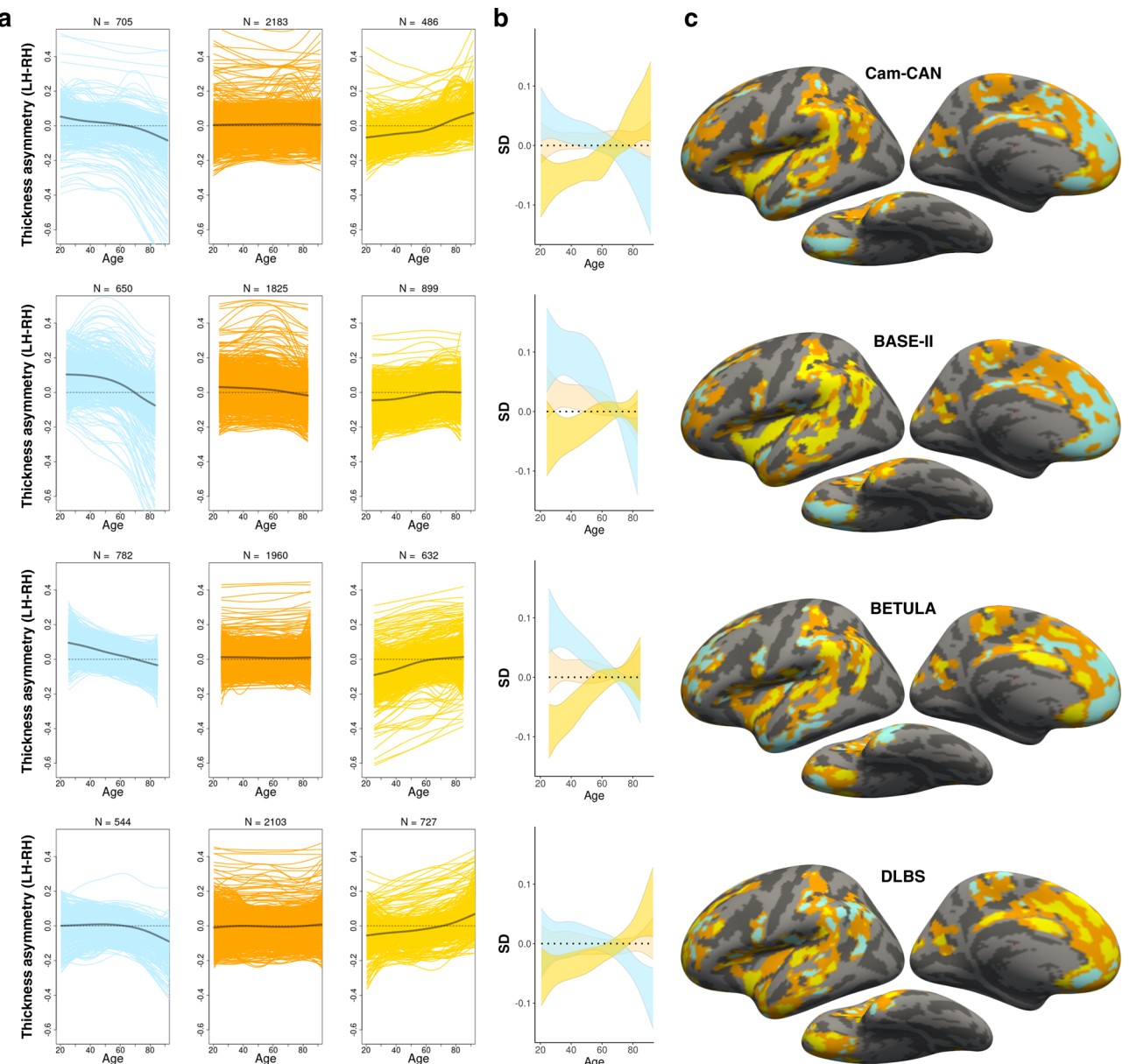

**Fig. 4 Replication. a** PAM clustering solutions of asymmetry trajectories in the four replication samples (depicted row-wise). Each colored line represents a vertex (N = number of vertices within each solution). Mean trajectories are in gray. In three out of four samples, the clustering solutions were highly similar to the discovery sample (cf. Fig. 3 and Supplementary Fig. 5). **b** Standard deviations (SD) of the asymmetry trajectories for the clustering solutions. **c** Solutions mapped on the surface. All trajectories were fitted using GAMMs. LH left hemisphere, RH right hemisphere.

Imaging Biomarkers and Lifestyle Study of Aging (AIBL; longitudinal observations only) to define longitudinal groups of healthy aging and AD individuals (see Fig. 5a and "Methods"), and ran LME analyses for each of the eight ROI's testing for group differences in asymmetry change over time. AD individuals were found to exhibit a steeper decline of (leftward) thickness asymmetry over time compared to cognitively healthy controls in most frontal cortical regions, and in anterior temporal cortex (Fig. 5b; see Supplementary Table 6 for full results). Whether or not an ROI showed opposite asymmetry in AD at baseline (i.e., thicker RH due to LH depletion) corresponded with whether the ROI lifespan trajectories showed opposite asymmetry in older age (cf. Fig. 3d; superior mPFC and orbitofrontal ROI's). Overall, the analyses confirmed that frontal and temporal patterns of asymmetry change across healthy adult life are accelerated in AD.

## Discussion

We discovered that the cerebral cortex thins asymmetrically across adult life. Age changes in asymmetry almost invariably reflected progressive asymmetry loss and hence were precipitated by faster thinning of the (previously) thicker homotopic hemisphere. In four out of five longitudinal adult lifespan samples the cortex clustered into a highly reproducible pattern that described age trajectories of asymmetry topologically across the cortex. This mapped onto a general anterior–posterior pattern of left–right thickness asymmetry and revealed loss of both leftward and rightward asymmetry on a similar time scale across adult life, suggesting system-wide loss of asymmetry in aging. Finally, we found that frontal and temporal regions vulnerable to asymmetry loss in healthy aging exhibit accelerated asymmetry change in AD.

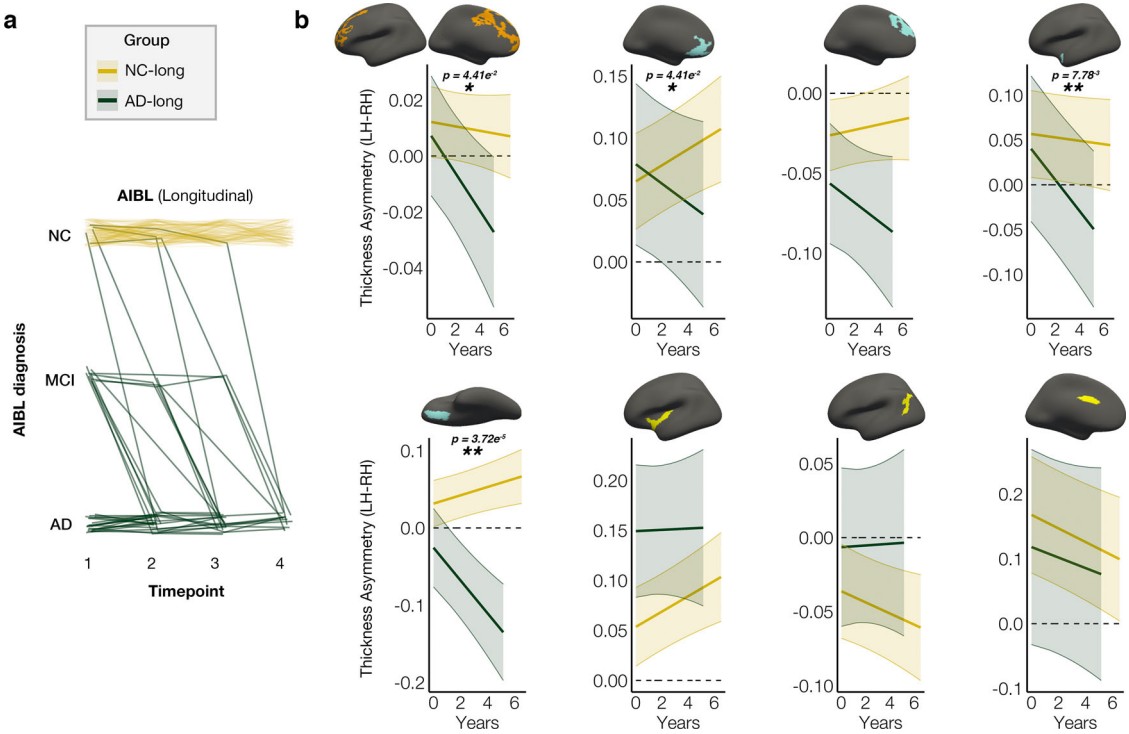

**Fig. 5 Longitudinal AD analysis.** Thickness asymmetry change in AD versus healthy aging in the AIBL clinical dementia sample. **a** *x*-axis denotes study timepoints. Single-timepoint diagnoses (*y*-axis; NC normal controls, MCI mild cognitive impairment, AD Alzheimer's disease) were used to define two longitudinal Groups of AD and NC individuals. Each line represents a subject and the color denotes longitudinal group membership (AD-long, in green; NC-long, in gold). AD-long individuals were diagnosed with AD by their final timepoint, whereas NC-long individuals were classified as healthy at every timepoint. Note that single-timepoint MCI diagnoses were considered only for the purpose of defining the longitudinal AD group (see "Methods"). **b** LME interaction between Group × Time (years) since baseline measurement upon asymmetry in clustering-derived ROI's (AD-long, $N = 41$, obs = 110; NC-long, $N = 128$, obs = 435; two-sided test). Colored lines depict mean thickness asymmetry (LH–RH) per group and ribbons depict 95% confidence intervals. Accelerated change in asymmetry was observed in the AD group in frontal and anterior temporal cortical ROI's. Results were corrected for multiple comparisons using FDR. Significant FDR-corrected *p* values are shown on plots (*$p$(FDR) < 0.05; ** <0.01; see Supplementary Table 6). LH left hemisphere, RH right hemisphere.

As we demonstrated, regional changes in cortical thickness asymmetry are an overlooked yet important feature of normal brain aging that is both shared by and accelerated in neurodegenerative AD. Across multiple cohorts, loss of leftward asymmetry over healthy adult life was particularly pronounced in frontal and anterior temporal cortex, and loss of rightward asymmetry was evident in insular and lateral temporoparietal regions. The average temporal dynamics of asymmetry loss were also markedly preserved across samples, with gradual asymmetry decline in early adulthood, and accelerated decline from age ~60 onwards. These trajectories applied to asymmetry fit well with general thinning trajectories known to show accelerated decline from age 60 onwards[21,38], though we note that individual regional trajectories did not always conform to the average time course. Regions we observed changing in asymmetry were predominantly located in higher-order association cortex, which has been found to exhibit less average interhemispheric connectivity[39] in favor of more lateralized within-hemisphere interactions[11]—supporting notions of hemispheric specialization[10,13]. Speculatively, our observation that thicker homotopic cortex thins faster may be in line with contemporary evidence indicating that age pathologies propagate transneuronally though synaptic connections[40,41] which may precipitate faster degeneration of the hemisphere with more cortico-cortical connections. As evidence suggests cortical thinning partly reflects synaptic loss[38,42,43], critical questions for future research concern the underlying neurobiological mechanisms, exact functional consequences, and indeed whether the principle that thicker cortex thins faster only applies to

homotopy. And while it remains unclear how the phenomenon may be reflected in age-related asymmetry differences in electrophysiological[44] and functional magnetic resonance signals[45,46], our findings may be in line with studies indicating decreased left lateralization of frontal resting-state networks with higher age[47]. Regardless, our results illustrate a hitherto undescribed phenomenon that fits within the well-established dedifferentiation view of the aging brain[3–5,7]: the mirrored reduction of leftward and rightward asymmetry across adult life suggests a system-wide breakdown in the adaptive asymmetric organization of the cortex in aging.

An asymmetrically organized cortex leads to increased proximity of collaborating brain regions and intrahemispheric clustering of specialized networks[12–14]. The asymmetry loss observed here may reflect decline in the intrahemispheric network efficiency of higher-order regions which are more inhibited from cross-hemispheric interaction[11,39] via the corpus callosum[48,49]. Because higher-order regions may also be more prone to homotopic disconnection with age[50], future work assessing how age changes in thickness asymmetry relate to altered interhemispheric connectivity and callosal degeneration[51,52] may shed light on important structure–function relationships in brain asymmetry and aging. Recent large-scale meta-analyses found evidence for age effects on thickness asymmetry only in superior temporal cortex[8] and previous cross-sectional studies have yielded mixed results[15,18,19]. Thus, by highlighting asymmetry loss as a system-wide process in aging, we here substantially extend previous knowledge. Though the reasons for the discrepancy between our

results and those reported by ENIGMA[8] are unclear, one possible explanation could be that longitudinal age changes in asymmetry effects are weak relative to the large interindividual variation in hemispheric thickness estimates that exists at any age, which may preclude detection of aging effects even in cross-sectional designs with thousands of participants (see[53]). It is also possible this issue may be exacerbated through parcellation-based approaches that average data over large regions of cortex if effects conform poorly to the predefined anatomical boundaries. By contrast, our longitudinal approach enabled modeling subject-specific variance in asymmetry over time to more accurately assess aging effects. Thus, the present study highlights the advantage longitudinal aging studies hold when it comes to detecting age-associated brain changes[54], as well as the advantage of vertex-wise asymmetry approaches. Regardless, the results presented here fit with the view that brain systems subserving higher-level associative cognition in particular become less specialized and more disorganized in aging[55].

The reliable anterior–posterior pattern of general thickness asymmetry found here (see Supplementary Fig. 7) agrees with previous work[9,15], has recently been shown in global meta-analyses[8], and is compatible with reports of developing thickness asymmetry from birth[56]. Reproducibility across samples and emerging cross-study consensus also in early development suggests a genetic influence upon cortical thickness asymmetry[56], and genetic factors have recently been implicated in the dynamics of age-related cortical change across life[21]. One can speculate whether age-related asymmetry breakdown is a by-product of genetic factors encoding the asymmetric organization of cortex[57] supporting hemispheric specialization of function, and that this differentiation leads to downstream neural consequences in aging. Though anatomo-functional relationships are likely complex[58], our results suggest that cortical thickness asymmetry may constitute a viable anatomical marker for key aspects of human hemispheric specialization. For example, we found robust evidence that mPFC asymmetry is particularly vulnerable in aging. Although implicated in a wide variety of complex cognition, the role of mPFC in executive function and normal memory operations is well established[59,60], and deficits in these are considered hallmarks of aging[37]. Although future research is needed to assess the specific cognitive relevance of regional thickness asymmetries and structure–function change relationships, the profound asymmetry loss observed here raises the possibility that complex cognitive abilities susceptible to decline in most individuals with advancing age may at least be partly subserved by hemispherically specialized networks.

Indeed, we found that asymmetry change extended through the aging–neurodegeneration continuum. Specifically, we observed accelerated loss of LH frontal and temporal cortices above and beyond those observed during healthy aging, illustrating that gradual age-related changes in asymmetry are exacerbated in AD. This agrees with previous longitudinal evidence indicating faster LH neurodegeneration in AD[30], recent work hinting at systemic AD-related asymmetry loss[31], and the regional susceptibility of frontal and temporal cortices to AD pathology[23,24]. One can speculate whether and to what degree faster LH degradation tracks to an asymmetric presence of other AD biomarkers, such as neurofibrillary tangles, as patterns of cortical thinning in AD largely overlap with tau deposition[24,61]. Future research could also assess whether and how cortical asymmetry change in AD might co-occur with asymmetric neurodegeneration of subcortical brain structures vulnerable in AD[62]. Here, we show the notion that characteristic changes in cortical structure in AD are also observed, though to a minor degree, in cognitively normal aging can also be extended to include thickness asymmetry[23–25]. Hence, the present findings suggest a continuity of change

between healthy and pathological brain aging, and highlight the importance of lifespan perspectives for understanding the pathophysiology of AD.

The implication that thickness asymmetry is important for healthy brain function agrees with a recent meta-analysis confirming that subtly reduced asymmetry is a feature of neurodevelopmental disorders[16]. However, our results also indicate that individual differences in regional asymmetry change had little predictive value upon longitudinal cognitive scores across cognitively healthy adult life (neither memory nor fluid reasoning ability). Thus, loss of thickness asymmetry may be more relevant for advanced cognitive decline—such as that associated with AD—but less sensitive to individual differences in cognitive decline trajectories across the healthy adult lifespan. Still, future work might benefit from assessing individual variability in asymmetry decline using high-dimensional data embedding techniques such as latent class analysis or t-SNE. Although we note that alternative tests such as cognitive speed measures may be better suited than task accuracy for assessing asymmetry–cognition relationships, our results suggest that asymmetry change may be unlikely to be a sensitive marker for age-related decline at the individual level (similar conclusions have recently been drawn in relation to asymmetry in other contexts[63]). This is also evidenced by the fact that our large longitudinal sample sizes were needed to expose small-to-medium effects that nevertheless translated to consistent gradual changes in asymmetry across adult lifespan samples.

Some potential caveats should be considered. First, regional differences in intracortical myelination of deep cortical layers will affect MRI-based estimates of cortical thickness[64], and hence it is possible that thickness asymmetry could partly reflect anatomically important differences in cortical myelination between hemispheres. Second, the estimation of GAMM age trajectories will be affected by age-distribution because knot placement is based upon data density, which differed across discovery and replication samples. This may explain some heterogeneity in vertex-wise results between samples (Supplementary Fig. 6) and in the subsequent clustering of asymmetry trajectories, and also limits the interpretation and replication of exact timings (e.g., acceleration and inflection points) of asymmetry change across adult life. The GAMM approach implemented here enabled flexibly fitting nonlinear age effects with relaxed assumptions of the shape of the adult lifespan trajectories[65]. While GAMMs are generally robust to non-normal distributions[65], it should be noted that the trajectories will be interpolated across missing age ranges (i.e., BASE-II; Fig. 1b), and may be somewhat affected by relative underrepresentation of specific age groups. This may best be appreciated by apparent inflection points around mid-adulthood evident in the asymmetry trajectories in LCBC data (Fig. 3); mid-adulthood is somewhat underrepresented in this sample largely due to naturally occurring selection bias against this age group in longitudinal lifespan research (Fig. 1a). The fact that such inflection points were not consistently evident in the data from the other cohorts may suggest these were driven by data density, and should not be overinterpreted. Relatedly, survivor bias in longitudinal aging studies will also affect the estimation of lifespan trajectories, and this could be one candidate explanation for the lack of full replication in DLBS. That is, if DLBS is more biased to recruiting high-performing older adults, this could preclude replication of the lifespan trajectories, and varying degrees of age-related pathology in later life could explain some of the differences in the lifespan reduction of asymmetry observed across samples. This would also agree with our observation that individuals who either had or were later diagnosed with AD (and hence may have been classed as cognitively normal at one or more timepoints) exhibited a higher reversal of the group-average asymmetry pattern at baseline. Third, because GAMMs estimate trajectories as a

combination of longitudinal and cross-sectional effects, the inclusion of more timepoints and longer follow-up intervals will better approximate the true longitudinal trajectories. Fourth, to ensure clustering reliability, we excluded small clusters assumed to be more prone to noise. Yet we also observed consistent vertex-wise effects in smaller regions of cortex that violated this assumption (e.g., parahippocampal gyrus; Supplementary Fig. 6), and thus cannot exclude other, more focal changes in asymmetry potentially informative for cognitive decline in aging. Fifth, because the clustering distributes all vertices among a given number of solutions, the trajectories of some vertices may not fit well within a given solution despite statistically fitting best to that solution. Consequently, the cluster solutions largely inform about the average trends rather than the asymmetry trajectories of individual regions, which showed more heterogeneity in shape. This forcing may also explain some spatial heterogeneity in clustering across samples. Nevertheless, the clustering protocol—which was based solely on the age trajectories of asymmetry and was blind to the spatial location in cortex—allowed for a concise description and comparison of the average asymmetry trajectories evident in each aging cohort.

Brain asymmetry seems to have arisen under evolutionary pressure to optimize processing efficiency and is broadly thought to confer organizational advantages that benefit brain function. Here, we show that the asymmetric organization of higher-order cortical regions in young adulthood decays with advancing age, and this decay follows a simple general organizing principle: the thicker of the two homotopic cortices thins faster. This principle was highly reproducible across different cohorts and was significantly accentuated in AD patients. Overall, the present study may have unveiled the structural basis of a widely suggested system-wide decline in hemispheric specialization across the adult lifespan in brain systems subserving higher-order cognition, and found a potential continuation and acceleration of this decline in AD.

## Method
### Samples
*Discovery sample.* An adult lifespan sample comprising 2577 scans from 1084 healthy individuals aged 20.0 to 89.4 (703 females) was drawn from the LCBC database (Department of Psychology, University of Oslo). See Supplementary Methods for details. All individuals were screened via health and neuropsychological assessments. The following exclusion criteria were applied across LCBC studies: evidence of neurologic or psychiatric disorders, use of medication known to affect the central nervous system (CNS), history of disease/injury affecting CNS function, and MRI contraindications. Participants in LCBC studies were required to consider themselves right-handed. Additionally, the following inclusion criteria were defined here: <21 on the Beck Depression Inventory, and ≥25 on the Mini Mental Status Exam. All LCBC studies were approved by the Norwegian Regional Committee for Medical and Health Research Ethics, all complied with the relevant ethical regulations, and all participants provided written informed consent.

*Replication samples.* Four longitudinal adult lifespan data sets were selected to test reliability of the discovery sample findings. Three were derived from the Lifebrain consortium—a data-sharing initiative between major European lifespan cohorts[54]: the Cam-CAN study ($N = 634$; $N$ scans $= 898$)[66], the BASE-II ($N = 447$; $N$ scans $= 768$)[67], and the BETULA ($N = 310$; $N$ scans $= 480$) project[68]. The fourth was the DLBS ($N = 471$; $N$ scans $= 763$)[69]. Single-timepoint and longitudinal data (up to two timepoints) were sourced for each replication cohort (see Supplementary Tables 1 and 3 for sample demographics and MRI image parameters).

*AD sample.* Data were collected by the AIBL study group (see[70]). We used the single-timepoint AIBL diagnosis (normal controls (NC); mild cognitive impairment (MCI); AD) to identify two longitudinal groups based on final-timepoint diagnosis: NC-long consisted of subjects classed as NC at every timepoint; AD-long consisted of subjects classed as AD throughout or by the final timepoint (see Fig. 5a). Individuals reverting from a MCI or AD diagnosis at a later timepoint were not included. In all, 545 observations of 169 subjects were included in the sample (timepoint range = 2–4; see Supplementary Table 2).

**MRI acquisition.** Discovery sample (LCBC) data consisted of T1-weighted (T1w) magnetization prepared rapid gradient echo sequences collected on two scanners at Oslo University Hospital; a 1.5 Tesla (T) Avanto and a 3 T Skyra (Siemens Medical Solutions, Germany). The number of Avanto and Skyra T1w scans was 832 and 1745, respectively. MRI acquisition parameters for all samples are summarized in Supplementary Table 3.

**MRI preprocessing.** Cortical reconstruction of the anatomical T1w images was performed with FreeSurfer's longitudinal pipeline (v6.0.0, http://surfer.nmr.mgh.harvard.edu/wiki)[71] on the Colossus computing cluster at the University of Oslo. This fully automated pipeline yields a reconstructed surface map of cortical thickness estimates for each participant and timepoint using robust inverse consistent registration to an unbiased within-subject template[72] (see Supplementary Methods). A high-resolution symmetrical surface template (LH_Sym)[33] was used to resample the FreeSurfer-estimated cortical thickness maps of the LH and RH of each participant into a common analysis space. This procedure achieves precise vertex-wise alignment between cortical hemispheres (see Fig. 1c and Supplementary Fig. 1). LH_Sym was created from a composite of LH and RH surface models in a database enriched in left-handers: the BIL&GIN (https://www.gin.cnrs.fr/en/tools/lh-sym/)[33,34]. In symmetrical space, an 8 mm full-width-half-maximum Gaussian smooth-kernel was applied across the surface to the LH and RH thickness data.

**Asymmetry-change analyses.** The analysis pipeline is summarized in Fig. 1d. Thickness maps were analyzed vertex-wise using GAMMs implemented in R (v3.5.0; gamm4 package[73]). We used a factor-smooth GAMM interaction approach that allowed fitting a smooth Age trajectory per Hemisphere and testing the smooth Age × Hemisphere interaction. Hemisphere was additionally included as a fixed-effect, Sex and Scanner as covariates-of-no-interest, and a random subject intercept was included. To minimize overfitting the number of knots was constrained to be low ($k = 6$). Significance of the smooth Age × Hemisphere interaction was formally assessed by testing for the existence of a difference between the smooth term of Age across Hemispheres. FDR-correction controlling for positive dependency[35] was applied to the resulting Age × Hemisphere and Hemisphere (main effect) significance maps, and the maps were masked at $p(FDR) < 0.001$.

**Clustering of asymmetry age trajectories.** The linear predictor matrix of the GAMM was used to obtain asymmetry trajectories [s(LH-Age)-s(RH-Age)] underlying the interaction, computed as the difference between zero-centered hemispheric trajectories. After removing the smallest clusters (<300 mm²), we resampled significant asymmetry trajectories to a lower resolution template (fsaverage5), calculated a dissimilarity matrix (see Fig. 1d), and submitted it to PAM clustering (cluster R package[36]). Clustering solutions from two to seven were considered, and the optimal solution was determined by the mean silhouette width. The largest cortical regions within each clustering solution were considered as ROI's in post hoc analyses (13 initial ROIs). Individual-level LH and RH thickness maps were then resampled from the symmetrical space to fsaverage5, and mean thickness was extracted for each ROI. To describe the patterns underlying the smooth interaction, for each ROI we ran comparable factor-smooth GAMMs and computed LH and RH trajectories. We then reran these after removing outliers (both hemispheres; defined as residuals ≥ 6 SD from the fitted trajectory of either hemisphere), and plotted the fitted Age trajectories for each hemisphere. Relative change was calculated by scaling fitted trajectories by the prediction at the minimum age. Eight main ROI's were identified to be used in subsequent ROI-based analyses based upon size and degree of asymmetry (see Supplementary Fig. 2).

**Replication analysis.** The GAMM analysis and clustering protocol were repeated for each longitudinal aging cohort. To evaluate clustering similarity the clustering was constrained within the same set of vertices as the discovery sample. For each clustering solution, Dice similarity was computed between discovery and replication maps, and the mean Dice across all solutions was taken. Significance of the spatial overlap in clustering results between discovery and replication samples was assessed across 10,000 random permutations of the clustering solution maps, as implemented in Matlab (R2017a). Equivalent vertex-wise factor-smooth GAMM analyses were ran for each sample independently—minus the Scanner covariate (all were single site).

**Cognitive-change analysis.** We considered longitudinal scores on the California Verbal Learning Test (CVLT) and the Matrix Reasoning subtest of the Weschler Abbreviated Scale of Intelligence as proxies of memory and fluid reasoning ability change, respectively. We used a subset of the LCBC sample for which at least two timepoints (range = 2–4) of cognitive tests were available. The total number of observations for CVLT and matrices with concurrent scans was 783 ($N = 312$) and 788 ($N = 314$), respectively (mean $N$ timepoints = 2.7; mean follow-up interval = 3.2 years; max interval = 10.9 years). For matrices, we used the raw scores. For CVLT, we used the first principal component across z-transformed raw scores on the learning, immediate, and delayed free-recall subtests. Next, for each of the eight main ROI's we ran GAMMs predicting each cognitive measure with the following

smooth terms: asymmetry (LH–RH), mean thickness (across hemispheres), and Age (Sex, Scanner, Test Version controlled; random subject intercept).

**Longitudinal AD analysis**. For AIBL data (April 28, 2015 release) we calculated thickness asymmetry (LH–RH) on LH_Sym, downsampled to fsaverage5, and extracted from clustering-derived ROIs. For each ROI, we ran LME's with the factors Group and Time (from baseline), and assessed the Group × Time interaction to test for group differences in asymmetry change. Age (at baseline), Sex, and Scanner were included as covariates, and a random subject intercept was included. As only year-of-birth data are available with AIBL, Age was estimated by randomly jittering the interval between the halfway date in the year-of-birth and test date.

**Reporting summary**. Further information on research design is available in the Nature Research Reporting Summary linked to this article.

## Data availability

All summary-level surface maps supporting the results are available on the Open Science Framework (OSF; DOI 10.17605/OSF.IO/XD7CF). These data can be used to reproduce all cohort-specific clustering analyses. The raw MRI data may be available upon reasonable request, given appropriate ethical, data protection, and data-sharing agreements. Requests for the raw MRI data can be submitted to the relevant principal investigator of each data contributing study[54] (https://www.lifebrain.uio.no/). Contact details are provided in Supplementary Note 4. Individual-level data availability for some of the samples is restricted as participants have not consented to publicly share their data, and different restrictions apply to different samples. LH_Sym is available at https://www.gin.cnrs.fr/en/tools/lh-sym/. AIBL data are available at https://aibl.csiro.au/research/support/ pending application approval and compliance with the data usage agreement.

## Code availability

All preprocessing and analysis code (together with example simulated data for the main analysis) is available at https://github.com/jamesmroe/AgeSym.

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

## Acknowledgements

The authors are indebted to Fabrice Crivello, Sophie Maingault, Nathalie Tzourio-Mazoyer, and Bernard Mazoyer for their generosity in sharing the surface template created in the BIL&GIN; a unique database enriched in left-handed individuals for the study of human brain lateralization. The Lifebrain project is funded by the EU Horizon 2020 Grant: "Healthy minds 0–100 years: Optimising the use of European brain imaging cohorts (Lifebrain)." Grant agreement number: 732592. In addition, the different sub-studies are supported by different sources: LCBC: The European Research Council under grant agreements 283634, 725025 (to A.M.F.), and 313440 (to K.B.W.), and the Norwegian Research Council (to A.M.F. and K.B.W.) under grant agreement 249931 (TOPPFORSK), The National Association for Public Health's dementia research program, Norway (to A.M.F). Some of the data used in the preparation of this article were obtained from the Australian Imaging Biomarkers and Lifestyle Flagship Study of Ageing (AIBL) funded by the Commonwealth Scientific and Industrial Research Organisation (CSIRO), which was made available at the ADNI database (www.loni.usc.edu/ADNI). The AIBL researchers contributed data but did not participate in analysis or writing of this report.

## Author contributions

J.M.R., R.W., D.V.-P., and A.M.F. conceived the study; J.M.R. and D.V.-P. performed analysis; Ø.S. contributed statistical expertise; A.M.M. managed cross-site data and relationships; R.A.K. and E.K. managed/procured/transferred Cam-CAN data; U.L. and S.K. managed/procured/transferred BASE-II data; D.C.P., M.M.R., and H.A.G. managed/procured/transferred DLBS data; L.N. and S.P. managed/procured/transferred Betula data; R.W., D.V.-P., Ø.S., A.M.B., S.D., R.A.K., E.K., S.K., U.L., A.M.M., D.C.P., S.P., M.M.R., K.B.W., and A.M.F. helped revise paper. R.W. and D.V.-P. provided supervision and reviewed/edited paper drafts; J.M.R. created figures and wrote the paper, with input from all authors.

## Competing interests

The authors declare no competing interests.

## Additional information

# The Australian Imaging Biomarkers and Lifestyle Flagship Study of Ageing

Colin L. Masters[9], Ashley I. Bush[9], Christopher Fowler[9], David Darby[9], Kelly Pertile[9], Carolina Restrepo[9], Blaine Roberts[9], Jo Robertson[9], Rebecca Rumble[9], Tim Ryan[9], Steven Collins[9], Christine Thai[9], Brett Trounson[9], Kate Lennon[9], Qiao-Xin Li[9], Fernanda Yevenes Ugarte[9], Irene Volitakis[9], Michael Vovos[9], Rob Williams[9], Jenalle Baker[9], Alyce Russell[10], Madeline Peretti[10], Lidija Milicic[10], Lucy Lim[11], Mark Rodrigues[11], Kevin Taddei[11], Tania Taddei[11], Eugene Hone[11], Florence Lim[11], Shane Fernandez[11], Stephanie Rainey-Smith[11], Steve Pedrini[11], Ralph Martins[11], James Doecke[12], Pierrick Bourgeat[12], Jurgen Fripp[12], Simon Gibson[12], Hugo Leroux[12], David Hanson[12], Vincent Dore[13], Ping Zhang[13], Samantha Burnham[13], Christopher C. Rowe[14], Victor L. Villemagne[14], Paul Yates[14], Sveltana Bozin Pejoska[14], Gareth Jones[14], David Ames[15], Elizabeth Cyarto[15], Nicola Lautenschlager[15], Kevin Barnham[16], Lesley Cheng[16], Andy Hill[16], Neil Killeen[17], Paul Maruff[18], Brendan Silbert[19], Belinda Brown[20], Harmid Sohrabi[20], Greg Savage[21] & Michael Vacher[22]

[9]The Florey Institute, The University of Melbourne, Parkville, VIC, Australia. [10]Collaborative Genomics Group, Centre of Excellence for Alzheimer's Disease Research and Care, School of Medical and Health Sciences, Edith Cowan University, Joondalup, WA, Australia. [11]School of Medical and Health Sciences, Edith Cowan University, Joondalup, WA, Australia. [12]CSIRO, Herston, QLD, Australia. [13]CSIRO, Melbourne, VIC, Australia. [14]Department of Molecular Imaging, Austin Health, Heidelberg, VIC, Australia. [15]National Ageing Research Institute, Parkville, VIC, Australia. [16]Bio21 Institute of Molecular Science and Biotechnology, The University of Melbourne, Parkville, VIC, Australia. [17]The University of Melbourne, Parkville, VIC, Australia. [18]Cogstate Ltd., Melbourne, VIC, Australia. [19]St. Vincent Hospital, Fitzroy, VIC, Australia. [20]Department of Exercise Science, College of Science, Health, Engineering and Education, Murdoch University, Murdoch, WA, Australia. [21]Department of Psychology, Macquarie University, Sydney, NSW, Australia. [22]CSIRO, Floreat, WA, Australia.

