## [Peer Review File · Nature Communications]

Reviewers' Comments:

Reviewer #1:

Remarks to the Author:

The authors present a convincing body of evidence for asymmetric age-related thinning of the cerebral cortex based on five longitudinal datasets, and its acceleration in Alzheimer's Disease in an additional dataset. The paper is clear and transparent. The findings support the authors' view that there is 'a system-wide dedifferentiation of the adaptive asymmetric organization of heteromodal cortex in aging and AD' at least in terms of cortical thickness, although they found no evidence that asymmetry change was related to cognitive performance change within healthy people.

Some points could be further clarified or explored:

1. In the Introduction the authors attribute conflicting results in previous literature to small sample sizes or not allowing non-linear effects of age. However, they also note that a larger study by the ENIGMA consortium (Kong et al 2018) found no effects of age on frontal asymmetry, which is discrepant with their findings. In addition, in the Results they note that 'An equivalent analysis with a Linear Mixed-Effect (LME) model showed comparable results' to non-linear analysis. Then it seems that the discrepancy with ENIGMA cannot be explained by either small sample size or the linear versus non-linear issue. If I understand the author's results correctly, one would expect the ENIGMA study to have found linear age-related effects consistent with those reported by the authors. Can the authors discuss other possible explanations for this discrepancy, and also adapt the introduction to consider additional possible explanations for discrepancies in general? E.g. the regional approach versus vertex-based approach (although it looks like the vertex-based results presented by the authors should also be detectable using a region-based approach, at least for medial frontal regions). Or do they think that ENIGMA's cross-sectional, meta-analytic approach was not suited to detect age-related changes? I am not sure since some ENIGMA datasets also had large age ranges, and the changes reported here by the authors would likely be visible as average age-related differences in large cross-sectional data, with age ranges spanning decades. I see that there is also a finding that matches ENIGMA, where the authors note that 'The main effect of Hemisphere (i.e. irrespective of Age) revealed a clear anterior-posterior pattern of general thickness asymmetry in adult life' (page 4). Here the authors can note that this is consistent with the previous report.

2. I have not worked with Generalized Additive Mixed Models, and am not sure how robust these are for the bimodal-age data used for the main discovery sample (see Fig 1a). The authors note some issues related to this in the Discussion, and the results are broadly consistent across their replication datasets, so I am not too concerned. They could more explicitly acknowledge the bimodal age distribution of the main sample in the Discussion, and how they think this might have influenced the findings from that sample.

3. Figure 1d, the age change trajectories for separate left and right measures switch from positive to negative at around age 50 or higher. Presumably this is due to zero-centering, but is a bit confusing for visualization since cortical thickness starts to decrease much earlier in life than 50 years of age. An explanatory note could be added in the figure legend.

4. In the text of the Results section for longitudinal AD analysis, please add the sample size to help the reader.

5. Other issues are appropriately dealt with in the analysis or discussed as limitations. For example the optimal clustering solution does not seem greatly superior to other solutions (Fig 1e), but the authors show evidence that the results are robust to this. They also discuss that asymmetry change was not related to cognitive performance change, except insofar as the AD dataset showed more marked asymmetry changes.

Clyde Francks

Reviewer #2:

Remarks to the Author:

The two hemispheres of the brain work in cooperation. Here, in a vast dataset James M. Roe et al. demonstrate that the balance between the hemispheres derived from cortical thickness estimates change with ageing, and is accelerated when combined with a brain pathology (i.e. Alzheimer disease).

The present work is excellent, and this is an exquisite contribution to the field.

I have a few comments.

1/ Cortical thickness derived from T1w is often biased by the myelination of the first layer of the cortex (Zilles and Amunts – Anatomical bases for functional specialisation). This should be discussed/ acknowledged in the manuscript.

2/ I did not find any data availability statement in the manuscript. I would recommend indicating how can other authors use these data to reproduce/build on the findings reported.

3/ Additional analyses may include high dimensional data embedding (e.g. TSNE) in order to see whether subgroups emerge in the dataset.

Response to the reviewers' comments.

Note that to comply with the journal guidelines, the manuscript title now reads "*Asymmetric thinning of the cerebral cortex across the adult lifespan is accelerated in Alzheimer's Disease*" (see page 5 of this document for additional minor changes)

Our responses are in green caption under each comment, manuscript text is italicized and denoted by page numbers and quotation marks, and changes are highlighted in blue (as in the revised manuscript).

General response: We would like to express our sincere gratitude to both reviewers for their insightful and encouraging comments. Our point-by-point response is detailed below.

Reviewer #1 (Remarks to the Author):

1. In the Introduction the authors attribute conflicting results in previous literature to small sample sizes or not allowing non-linear effects of age. However, they also note that a larger study by the ENIGMA consortium (Kong et al 2018) found no effects of age on frontal asymmetry, which is discrepant with their findings. In addition, in the Results they note that 'An equivalent analysis with a Linear Mixed-Effect (LME) model showed comparable results' to non-linear analysis. Then it seems that the discrepancy with ENIGMA cannot be explained by either small sample size or the linear versus non-linear issue. If I understand the author's results correctly, one would expect the ENIGMA study to have found linear age-related effects consistent with those reported by the authors. Can the authors discuss other possible explanations for this discrepancy and adapt the introduction to consider additional possible explanations for discrepancies in general? E.g. the regional approach versus vertex-based approach (although it looks like the vertex-based results presented by the authors should also be detectable using a region-based approach, at least for medial frontal regions). Or do they think that ENIGMA's cross-sectional, meta-analytic approach was not suited to detect age-related changes? I am not sure since some ENIGMA datasets also had large age ranges, and the changes reported here by the authors would likely be visible as average age-related differences in large cross-sectional data, with age ranges spanning decades.

Response: We agree with the reviewer's interpretation of our results: our non-linear approach was not strictly necessary to reveal the age-changes, as 96% of significant vertices with the GAMM approach were significant with longitudinal linear mixed models (shown in SI). We also agree with the reviewer that it is not clear why the ENIGMA meta-analysis did not detect age-effects, particularly in mPFC. One explanation may be that the longitudinal effect sizes of age-changes in thickness asymmetry are small compared to the larger individual differences in hemispheric cortical thickness estimates evident at any age. In this case, longitudinal analyses control a significant amount of error variance by binding individual differences to the random effects, making the analysis more sensitive for the discovery of smaller effects. This interpretation would be in line with recent Lifebrain analyses, which found that small longitudinal effects were not detected with large cross-sectional data, neither with N=3000 (Lifebrain) nor N=20,755 (UK Biobank)¹. This suggests that small longitudinal interactions associated with age-related brain changes may indeed be difficult to detect in large cross-sectional data, which we believe is one candidate explanation for the discrepancy between our results and ENIGMA's. We now address this specifically in the Discussion.

In addition, we agree that differences in the region-wise versus vertex-wise approach the reviewer mentions are likely important, as the vertex-wise structure of thickness asymmetry may not conform well to the Desikan-Killiany atlas. We have added the following sentences in the Introduction and Discussion to reflect this:

P2. “Conflicting results may be partly ascribable to differences in sample size and test power, and linear modelling of cross-sectional age effects across developmental and aging samples known to follow non-linear brain trajectories over time^{2,3}. Methodological differences such as whether age-effects upon thickness asymmetry were probed regionally^{4,5} or vertex-wise across the cortex^{6,7} may also help explain divergent results. Longitudinal lifespan studies examining intra-individual asymmetry change have the potential to resolve inconsistencies and capture non-linear asymmetry-change trajectories, but are currently absent from the literature.”

P11. “Recent large-scale meta-analyses found evidence for age effects on thickness asymmetry only in superior temporal cortex⁴ and previous cross-sectional studies have yielded mixed results⁵⁻⁷. Thus, by highlighting asymmetry-loss as a system-wide process in aging we here substantially extend previous knowledge. Though the reasons for the discrepancy between our results and those reported by ENIGMA⁴ are unclear, one possible explanation could be that longitudinal age-changes in asymmetry effects are weak relative to the large inter-individual variation in hemispheric thickness estimates that exists at any age, which may preclude detection of aging effects even in cross-sectional designs with thousands of participants (see¹). It is also possible this issue may be exacerbated through parcellation-based approaches that average data over large regions of cortex if effects conform poorly to the predefined anatomical boundaries. By contrast, our longitudinal approach enabled modelling subject-specific variance in asymmetry over time to more accurately assess aging effects. Thus, the present study highlights the advantage longitudinal aging studies hold when it comes to detecting age-associated brain changes⁸, as well as the advantage of vertex-wise asymmetry approaches.”

2. I see that there is also a finding that matches ENIGMA, where the authors note that ‘The main effect of Hemisphere (i.e. irrespective of Age) revealed a clear anterior-posterior pattern of general thickness asymmetry in adult life’ (page 4). Here the authors can note that this is consistent with the previous report.

Response: We have amended the results accordingly:

P4. “The main effect of Hemisphere (i.e. irrespective of Age) revealed a clear anterior-posterior pattern of general thickness asymmetry in adult life (Fig. 2A), consistent with a recent meta-analysis⁴”

2. I have not worked with Generalized Additive Mixed Models, and am not sure how robust these are for the bimodal-age data used for the main discovery sample (see Fig 1a). The authors note some issues related to this in the Discussion, and the results are broadly consistent across their replication datasets, so I am not too concerned. They could more explicitly acknowledge the bimodal age distribution of the main sample in the Discussion, and how they think this might have influenced the findings from that sample.

Response: Thank you for this comment. GAMMs are well-suited to handle predictive variables (e.g. age) with an unequal distribution, placing several “knots” along the variable continuum to minimize this type of bias. As such, the asymmetry trajectories are relatively robust to the somewhat bimodal age-distribution in LCBC. However, we agree with the reviewer that the evident inflection points around mid-adulthood in LCBC data do suggest a slight influence of the age-distribution on the fitted trajectories. We have added the following sentences to the limitations/Discussion to explicitly address this and provide additional information on the GAMM approach in general:

P12. “Second, the estimation of GAMM age-trajectories will be affected by age-distribution because knot placement is based upon data density, which differed across discovery and replication samples. This may explain some heterogeneity in vertex-wise results between samples (SI Fig.6) and in the subsequent clustering of asymmetry trajectories, and also limits the interpretation and replication of exact timings (e.g. acceleration and inflection points) of asymmetry change across adult life. The GAMM approach implemented here enabled flexibly fitting non-linear age effects with relaxed assumptions of the shape of the adult lifespan trajectories⁹. While GAMMs are generally robust to non-normal distributions⁹, it should be noted that the trajectories will be interpolated across missing age-ranges (i.e. BASE-II; Fig 1B), and may be somewhat affected by relative

underrepresentation of specific age-groups. This may best be appreciated by apparent inflection points around mid-adulthood evident in the asymmetry trajectories in LCBC data (Fig. 3); mid-adulthood is somewhat underrepresented in this sample largely due to naturally occurring selection bias against this age-group in longitudinal lifespan research (Fig. 1A.). The fact that such inflection points were not consistently evident in the data from the other cohorts may suggest these were driven by data density, and should not be overinterpreted.”

3. Figure 1d, the age change trajectories for separate left and right measures switch from positive to negative at around age 50 or higher. Presumably this is due to zero-centering, but is a bit confusing for visualization since cortical thickness starts to decrease much earlier in life than 50 years of age. An explanatory note could be added in the figure legend.

Response: We apologize that Fig. 1d was unclear. This figure shows the absolute age trajectories of the cortical thickness of each hemisphere (in mm) after each has been demeaned (i.e. zero-centered). We refer to zero-centering to be consistent with the GAMM literature. We have made the following changes to the legend to make this clear:

P4. “d) Visualization of asymmetry analysis at an example vertex (black circle). GAMMs were used to compute the zero-centered (i.e. demeaned) age-trajectories of the left [s(LH-Age)] and right [s(RH-Age)] hemisphere at every vertex. Blue and yellow lines indicate the smooth age trajectory of cortical thickness for the left and right hemisphere, respectively, after each has been demeaned. The difference between these (green line) depicts the asymmetry trajectory [s(LH-Age)-s(RH-Age)] that was used for assessment of age-change in asymmetry, equivalent to the GAMM interaction term.”

4. In the text of the Results section for longitudinal AD analysis, please add the sample size to help the reader.

Response: The legend now includes:

P10. “AD-long individuals (N = 41, obs = 110) were diagnosed with AD by their final timepoint whereas NC-long individuals (N = 128, obs = 435) were classified as healthy at every timepoint.”

Other issues are appropriately dealt with in the analysis or discussed as limitations. For example the optimal clustering solution does not seem greatly superior to other solutions (Fig 1e), but the authors show evidence that the results are robust to this. They also discuss that asymmetry change was not related to cognitive performance change, except insofar as the AD dataset showed more marked asymmetry changes.

Reviewer #2 (Remarks to the Author):

1/ Cortical thickness derived from T1w is often biased by the myelination of the first layer of the cortex (Zilles and Amunts – Anatomical bases for functional specialisation). This should be discussed/ acknowledged in the manuscript.

We agree that this is an issue to address in the context of our study. We have added the following sentence to explicitly address this at the outset of the limitations section in the Discussion:

P12. “Some potential caveats should be considered. First, regional differences in intracortical myelination of deep cortical layers will affect MRI-based estimates of cortical thickness¹⁰, and hence it is possible that thickness asymmetry could partly reflect anatomically important differences in cortical myelination between hemispheres.”

2/ I did not find any data availability statement in the manuscript. I would recommend indicating how can other authors use these data to reproduce/build on the findings reported.

Response: For ethical reasons, access to all individual-level data associated with this work is unfortunately restricted, as depending on the dataset (e.g., LCBC), participants have not consented to publicly share their data. Beyond this, we will make every effort to be transparent, sharing all summary-level maps of effects in SI and all associated code. We have added the following Data availability statement and Code availability statement to comply with the journal guidelines:

*P17. “Data availability
The raw MRI data supporting the results of the current study may be available upon reasonable request, given appropriate ethical, data protection approvals and data sharing agreements. Requests for the raw MRI data can be submitted to the relevant principal investigator of each study⁸ (<https://www.lifebrain.uio.no/>). Contact information can be obtained from the corresponding author. Individual-level data availability is restricted as participants have not consented to publicly share their data, and different restrictions apply to different samples. All summary-level surface maps supporting the results of the current study are available in supporting information. “LH Sym” is openly available at <https://www.gin.cnrs.fr/en/tools/lh-sym/>. AIBL data is open. Approval must be requested at <https://aibl.csiro.au/research/support/>.*

*Code availability
All preprocessing and analysis code is available at <https://github.com/jamesmroe/AgeSym>”*

3/ Additional analyses may include high dimensional data embedding (e.g. TSNE) in order to see whether subgroups emerge in the dataset.

Response: Thank you very much for the suggestion. We agree that identifying subgroups is an interesting possibility for further exploration of the data. Since the current paper is already complex and analysis-rich, we will prefer not to add additional analyses that would extend the manuscript significantly, but acknowledge the reviewer’s point as a good idea for future studies. We now address this point directly in the Discussion section of the manuscript:

P12. “Thus, loss of thickness asymmetry may be more relevant for advanced cognitive decline – such as that associated with AD – but less sensitive to individual differences in cognitive decline trajectories across the healthy adult lifespan. Still, future work might benefit from assessing individual variability in asymmetry decline using high-dimensional data embedding techniques such as latent class analysis or t-SNE.”

Additional changes made to comply with journal guidelines include Data/Code availability statements (see Reviewer 2 response) and the following text:

P2. “Here, in an initial longitudinal adult lifespan discovery sample (LCBC; see Fig. 1A; Age-range = 20-89; total observations = 2577; longitudinal observations = 1851; mean follow-up interval = 2.7 years), we show that the cerebral cortex thins asymmetrically in aging

..... Finally, we predicted that similar changes in regional thickness asymmetry will occur faster in AD, and show this using a longitudinal clinical dementia dataset.”

P14. All LCBC studies received approval from the Regional Ethical Committee of South Norway, and all participants provided written informed consent.

Figure caption titles:

Figure 1: **Samples and analysis method.**

Figure 2: **Vertex-wise GAMM effects.**

Figure 3: **Clustered trajectories.**

Figure 4: **Replication.**

Figure 6: **Longitudinal AD analysis**

References

1. Fjell, A. M. *et al.* Self-reported sleep relates to hippocampal atrophy across the adult lifespan - results from the Lifebrain consortium. *Sleep* 1–15 (2019). doi:10.1093/sleep/zsz280
2. Fjell, A. M. *et al.* Development and aging of cortical thickness correspond to genetic organization patterns. *Proc. Natl. Acad. Sci. U. S. A.* **112**, 15462–15467 (2015).
3. Tamnes, C. K. *et al.* Development of the Cerebral Cortex across Adolescence: A Multisample Study of Inter-Related Longitudinal Changes in Cortical Volume, Surface Area, and Thickness. *J. Neurosci.* **37**, 3402–3412 (2017).
4. Kong, X.-Z. *et al.* Mapping cortical brain asymmetry in 17,141 healthy individuals worldwide via the ENIGMA Consortium. *Proc. Natl. Acad. Sci.* **115**, E5154–E5163 (2018).
5. Long, X., Zhang, L., Liao, W., Jiang, C. & Qiu, B. Distinct laterality alterations distinguish mild cognitive impairment and Alzheimer’s disease from healthy aging: Statistical parametric mapping with high resolution MRI. *Hum. Brain Mapp.* **34**, 3400–3410 (2013).
6. Plessen, K. J., Hugdahl, K., Bansal, R., Hao, X. & Peterson, B. S. Sex, age, and cognitive correlates of asymmetries in thickness of the cortical mantle across the life span. *J Neurosci* **34**, 6294–6302 (2014).
7. Zhou, D., Lebel, C., Evans, A. & Beaulieu, C. Cortical thickness asymmetry from childhood to older adulthood. *Neuroimage* **83**, 66–74 (2013).
8. Walhovd, K. B. *et al.* Healthy minds 0–100 years: Optimising the use of European brain imaging cohorts (“Lifebrain”). *Eur. Psychiatry* **50**, 47–56 (2018).
9. Sørensen, Ø., Walhovd, K. B. & Fjell, A. M. A Recipe for Accurate Estimation of Lifespan Brain Trajectories, Distinguishing Longitudinal and Cohort Effects. (2020).
10. Natu, V. S. *et al.* Apparent thinning of human visual cortex during childhood is associated with myelination. *Proc. Natl. Acad. Sci.* **116**, 20750–20759 (2019).

Reviewers' Comments:

Reviewer #1:

Remarks to the Author:

Thank you for considering and addressing my comments.

Reviewer #2:

Remarks to the Author:

The reviewers appropriately replied to my requests. I have no further comments. This is an excellent paper.